# Xylem Sap Mycobiota in Grapevine Naturally Infected with *Xylella fastidiosa*: A Case Study: Interaction of *Xylella fastidiosa* with *Sclerotinia sclerotiorum*

**DOI:** 10.3390/plants14131976

**Published:** 2025-06-27

**Authors:** Analía Perelló, Antonia Romero-Munar, Sergio I. Martinez, Antonio Busquets, María Cañellas, Bárbara M. Quetglas, Rafael Bosch, Jaume Vadell, Catalina Cabot, Marga Gomila

**Affiliations:** 1UCA-FICA-CONICET, Facultad de Ingeniería y Ciencias Agrarias, Pontificia Universidad Católica Argentina, Av. Alicia Moreau de Justo 1300, CABA, Buenos Aires C1107AFD, Argentina; 2Department of Biology, University of the Balearic Islands, Cra. Valldemossa, km 7,5, 07122 Palma, Spain; antonia.romero@uib.cat (A.R.-M.); mbarrera@uib.es (M.C.); bquetglas@uib.es (B.M.Q.); r.bosch@uib.es (R.B.); jaume.vadell@uib.es (J.V.); marga.gomila@uib.es (M.G.); 3Centro de Investigaciones en Fitopatología, Consejo Nacional de Investigaciones Científicas y Técnicas, Facultad de Ciencias Agrarias y Forestales, Universidad Nacional de La Plata (CIDEFI-CONICET FCA y F-UNLP), Buenos Aires C1900, Argentina; sergio.martinez@agro.unlp.edu.ar; 4Scientific-Tecnhical Services, Universitat de les Illes Balears, Cra. Valldemossa, km 7,5, 07122 Palma, Spain; toni.busquets@uib.es

**Keywords:** endophytic mycobiota, grapevine, *Xylella fastidiosa*, cross-kingdom interactions, synergism, *Sclerotinia sclerotiorum*

## Abstract

Grapevine (*Vitis vinifera*) is a key crop in Mediterranean agriculture, now increasingly threatened by *Xylella fastidiosa* subsp. *Fastidiosa* (*Xff*), the causal agent of Pierce’s disease. This study investigated: (1) the diversity of culturable fungal endophytes in the xylem sap of naturally *Xff*-infected grapevines, and (2) the interaction between *Xff* and the pathogenic fungus *Sclerotinia sclerotiorum* identified in the sap. The xylem sap was collected from Cabernet Sauvignon vines in Mallorca, Spain, and fungal communities were characterized using culture-dependent methods. Both beneficial fungi (e.g., *Aureobasidium pullulans*, *Rhodotorula mucilaginosa*) and pathogenic species (e.g., *S. sclerotiorum*, *Cladosporium* sp., *Alternaria alternata*, and the *Phoma* complex) were isolated from both *Xff*-positive and *Xff*-negative plants, indicating similar community profiles. Although limited by small sample size, these findings offer preliminary evidence of complex ecological interactions between *Xff* and the xylem-associated mycobiota, with potential implications for grapevine health and disease development under varying environmental and management conditions. Further experiments under controlled conditions revealed that grapevines co-inoculated with *Xff* and *S. sclerotiorum* showed increased disease severity, suggesting a synergistic interaction. These preliminary results highlight the complex interplay between *Xff* and the fungal endophytic microbiome, which may modulate grapevine susceptibility depending on environmental and management conditions.

## 1. Introduction

Grapevine (*Vitis vinifera*) is one of the most widely cultivated fruit crops with a great economic impact on the agriculture industry. This crop is part of the landscape of the entire Mediterranean region where viticulture is of high economic relevance, whether for fruit fresh consumption, dried fruits or for wine production. In Mallorca, the history of wine dates back to the Romans who introduced the cultivation of grapevines to the island and ever since, wine has continued to be produced with varying degrees of success. A major recent challenge for viticulture in Mallorca was the detection in 2016 of *Xylella fastidiosa*, a quarantine pathogen in the European Union. Specifically, *Xylella fastidiosa* subsp. *Fastidiosa* (*Xff*) [1] is the causal agent of Pierce’s Disease (PD), one of the most destructive diseases for viticulture in affected regions and has led to substantial economic losses for the wine industry [2]. Alarmingly, climate change is expected to exacerbate PD outbreaks in the Mediterranean, where increasingly severe summers and milder winters may promote the bacterium’s spread [3].

*Xylella fastidiosa* is a gram-negative bacterium restricted to the xylem and known for its slow growth in culture. It is naturally transmitted by xylem-feeding insects, with transmission efficiency varying by vector species [4].

The bacterium overwinters in the xylem of the host plants as well as in weeds [5] and do not kill the hosts until later stages of its life cycle. Moreover, this causal agent can also live asymptomatically as an endophyte [2,4]. In susceptible grapevines, *Xff* causes xylem vessel occlusion reducing hydraulic conductivity [6]. The grapevine’s defense response includes the formation of tyloses, a mechanism also involved in combating grapevine trunk diseases (GTDs) [7].

Despite the importance of *Xff*, few studies have examined its interactions with the grapevine’s xylem sap endophytic community. Changes in the bacterial population and disease symptoms expression in *Xff*-infected grapevines were reported [8]. Moreover, different components of the microbiome in *Xff*-infected grapevines with antimicrobial activity and plant growth-promoting properties have been reported recently. Additionally, few reports on the potential synergic effects that the microbiome could exert on *Xff* virulence are available. For example, the presence of the endophytic N-fixing *Methylobacterium* increased the symptoms caused by *Xff* subsp. *Pauca* in *Citrus sinensis* is mentioned [9]. However, interactions between *Xff* and fungal pathogens remain poorly understood. Given the potential for co-infection, such interactions could significantly affect disease dynamics and epidemiology.

*Sclerotinia sclerotiorum* (Ss) is a devastating necrotrophic fungus affecting over 500 plant species worldwide [10]. It belongs to the *Sclerotiniaceae* family and is characterized by white cottony mycelium and melanized sclerotia, of which can remain viable in soil for up to 10 years [11]. Infection occurs through two germination pathways: carpogenic germination, where sclerotia release airborne ascospores that land on plant tissues, germinate, and penetrate using decaying material as a nutrient source [12], and myceliogenic germination, in which sclerotia germinate directly, forming hyphae that infect the plant stem base or produce new sclerotia in the absence of a host [13,14]. This fungus was reported on grapevines causing shoot blight in Chile [15,16,17,18]. Additionally, this fungus was reported among the endophytic mycobiota associated with *Vitis vinifera* in the Iberian Peninsula [19] but not as a pathogen causing visible symptoms on that crop. In 2021, grapevines cv. Callet growing in a commercial vineyard located northeast of the island of Mallorca showed severe symptoms of canker and shoot blight during spring and early summer, with a 70% incidence [20]. The presence of Ss was also confirmed as a component of the mycobiota of the xylem sap of the vineyard sampled in this current study.

Taking the above into consideration, due to the lack of information and data on grapevine endophytic communities in Mallorca, this work aimed to (1) preliminarily explore the diversity of the culturable fungal endophytes in grapevine xylem sap focusing on organisms that share ecological niches with *Xff* and may serve as biological control agents or co-occurring pathogens; (2) Investigate the interaction between *Xff* and the phytopathogenic fungus *S. sclerotiorum*, with the goal of understanding how these microorganisms may influence each other within grapevine hosts.

## 2. Results

### 2.1. Xylella fastidiosa subsp. Fastidiosa PCR Identification

The molecular identification of *Xylella fastidiosa* subsp. *Fastidiosa* (*Xff*) was successfully performed using real-time PCR (qPCR). Detection was carried out following two specific protocols described by Francis et al. (2008) [21], employing the primer pair XF-F/XF-R with the TaqMan probe XF-P, and the primer pair HL5/HL6 with a corresponding TaqMan probe. Amplifications were performed using a QuantStudio 3 Real-Time PCR System (Applied Biosystems). Both primer sets enabled efficient and specific amplification, confirming the presence of *Xff* in the analyzed samples.

### 2.2. Fungal Microbial Diversity Recovered from the Sap of Grapevine Plants

The fungal identification is shown in Figure 1. These results indicated the different groups of microbiota diversity in xylem sap of grapevines cv. Cabernet Sauvignon recovered. Most of the endophytic fungi recovered belonged to the Phylum Ascomycota with two main groups: yeast-like and filamentous (mycelial). These species can be considered to belong to different functional groups, true endophytes, beneficial saprophytes, opportunistic or not, and latent pathogens associated with the trunk’s diseases.

Regarding the first group, the yeast-like fungi, the morphocultural and microscopic analysis showed the presence of single-celled, spherical or elliptical spores, 3–15 µm in size, which could give rise to the formation of pseudohyphae (yeasts) when the sprouted cell does not separate from the mother cell. On Sabouraud agar, pale and opaque, mucilaginous colonies developed, with some species with characteristic pigments, although they were generally cream, pink or dark in color. Microscopically, most of the yeast species differed very little and thus physiological tests are necessary for their complete identification. Among them, *Aureobasidium pullulans*, *Rhodotorula mucilaginosa* and other yeast-like fungi that were recovered from the samples, stand out for their potential use in future biological control tests that could position them as promising antagonists to modulate the impact of GTDs.

The second group recovered, the filamentous fungi, belong to the Phylum Ascomycota. Some of them show only asexual reproduction and several types of vegetative spores without conidiomata. Among them, *Penicillium* spp. Complex, along with *P. chrysogenum,* were recovered from the sap of grapevines that had tested positive or negative for the occurrence of *Xff.* Moreover, some fungi of potential risk, specifically those previously mentioned as pathogens for grapevine like *Sclerotinia sclerotiorum*, *Cladosporium* sp and *Alternaria alternata* were also identified. Other cultivable fungi, such as the *Phoma* complex and the *Phomopsis/Diaporthe* complex—identified by their pycnidial dark conidia—were recovered from both *Xff*-positive and *Xff*-negative samples.

Depending on their presence in the xylem sap of plants that tested positive or negative for the occurrence of *Xff*, the differences in the fungal diversity complex recovered are shown in Figure 1. Qualitative–quantitative differences in the structure of the microbiota recovered were found to show a higher relative diversity in sap samples of plants that tested positive for the presence of *Xff* compared to those samples that tested negative to the bacteria. A total of 13 different groups/complexes of fungi were recovered; however, some of these, like *P. chrysogenum, A. alternata, Cladosporium* sp., only appeared in plants that had tested negative to the presence of *Xff*, while *S. sclerotiorum,* yeast-like, *Phoma complex* and *A. pullulans* appeared associated in plants that tested positive to the bacteria. The rest of the microorganism isolated were shared by both groups (*Aspergillus* sp., *Botryosphaeria* complex, Coelomycetes, *Penicillium* spp., *Phaeoacremonium*/*Phaeomoniella* sp., *Phomopsis/Diaporthe* complex, *Rhodotorula mucilaginosa*). Related to the *Dematiaceous* complex group, the morphocultural analysis has shown a great presence of Coelomycetes, with conidia formed in closed or partly closed fruiting structures with type pycnidial conidiomata e.g *Phoma* complex clade (Figure 1). Interestingly, in addition to those species belonging to the Dematiaceous complex identified here, the other taxonomical members of the Coelomycetes group like *Phomopsis/Diaporthe* complex and *Cladosporium* spp. Were identified by morphobiometrical features (Appendix A) and their identities need to be corroborated by molecular techniques.

### 2.3. Interaction Between Sclerotinia sclerotiorum (Ss) and Xylella fastidiosa (Xff) on Grapevine Plants Under Greenhouse-Controlled Conditions

After the artificial infection with *Ss*, the symptoms observed 7 days post inoculation (dpi) were necrosis in the stems, which evolved in an elongated and extended shape up and down from the initial point of infection, as the disease progressed. As a result of the necrotic lesion, many shoots broke and fell prematurely at 5 dpi. Other symptoms recorded were necrosis of the leaf petioles, epinasty and, as a consequence, leaf wilting of the compromised leaves in the shoots with symptoms (Figure 2 and Figure 3).

Significant differences were found between *Ss* and *Ss + Xff* treatments for the average number of infective lesions on stems from the initial point of inoculation. The co-inoculation *Ss + Xff* showed a higher number of lesions than the others treatments. A significant number of higher initial inoculation points developed in necrotic lesions in the combination *Ss* + *Xff*, in comparison with the *Ss* treatment (Figure 4A). Also, significant differences in the length of the necrotic lesions (16.88 cm vs. 46.05 cm) and the number of necrotic petioles/plant (18 vs. 27) between *Ss + Xff* and *Ss* were found (Figure 4B,C). The *Xff* control plants and healthy control without inoculation plants (Mock), did not register necrotic symptoms. It is important to highlight the greater disease intensity over time and faster progress found in the combination *Xff-Ss* compared to *Ss*.

In Figure 5, the biplot generated from principal component analysis (PCA) indicated that the first two principal components (F1 and F2) explain 98.14% of the total variability of the data, with a predominance of the first component (87.52%). This suggests that most of the variation in the data is captured by the F1 axis. The three variables studied (number of necrotic petioles, number of lesions on stems, and total length of the lesions) are closely correlated, since their arrows point in similar directions. These variables mainly contribute to the first component (F1), indicating that F1 summarizes the combined effect of these measurements. Moreover, the *Xff*-Ss treatment (plants with both infections) is strongly associated with high values in all variables analyzed, being in the same direction as the arrows. The *S. sclerotiorum* (Ss) treatment shows a moderate response, partially separating from the control and the plants with *Xylella fastidiosa* (*Xff*). Mock (control) and *Xff* (only *Xylella*) are in the left quadrant, indicating that they present low values in the measured variables. Thus, the biplot suggests that co-infection (*Xff*-Ss) generates a greater impact on the three parameters evaluated, while Mock plants and with *Xff* show low levels of necrosis. This shows a synergic effect between *Sclerotinia* and *Xylella* on the severity of symptoms (Figure 5).

The biplot displays the first two principal components, F1 (87.52%) and F2 (10.62%), which together explain 98.14% of the total variance in the dataset. Treatments are represented as blue dots and include *Xff*, *Xff-Ss*, *Ss,* and Mock groups. Red vectors indicate the disease severity variables: number of lesions on stems, necrotic stem length (cm), and number of necrotic petioles per plant. Treatments inoculated with *Xff-Ss* are positively associated with higher values of all three disease parameters, as indicated by their position along the direction of the vectors. In contrast, Mock and Ss treatments cluster near the origin, reflecting minimal or no disease symptoms. The PCA suggests that the *Xff-Ss* treatment caused the most severe symptoms, distinguishing it clearly from other treatments along PC1.

The graphic representation of the total accumulated disease measured as the area under the disease progress curve (AUDPC) is shown in Figure 6. Analysis of the data indicates a relatively higher accumulated disease—assessed as stem necrotic lesion length (cm) at three progressive observation times during the experiment—registered in the combination *Xff*-Ss (A) in comparison with the AUDPC in treatment Ss (B). Moreover, a faster velocity to increase in symptoms was shown in the first treatment (A).

Although the types of symptoms recorded were the same in both treatments (*Ss* and *Xff + Ss*), a significantly greater intensity of symptoms and aggressiveness of *Sclerotinia* stands out for plants previously infected with *Xff* (Figure 7).

Moreover, changes were detected in the stomatal conductance of plants affected by *Xff* in the presence of *Ss.* Our results showed a different response induced by the *Ss* and *Xff* with respect to control plants. According to our results, grapevine plants subjected to stress caused by *Ss* showed highly significant values of stomatal conductance (451 mmol m^−2^s^−1^) compared to the control plants (199 mmol m^−2^s^−1^). These high values were also reflected in the interaction of the fungus with *Xff* in concomitant infections (362,3 mmol m^−2^s^−1^). The lowest stomata opening value was induced by *Xff* (139 mmol m^−2^s^−1^) although without statistically significant differences with control plants. Regarding chlorophyll content, non-significant differences were found among treatments (Figure 8).

## 3. Discussion

*Vitis vinifera* hosts a complex of endophytic microorganisms that interact among themselves and within the plant. Those microorganisms can be beneficial, neutral or pathogenic to the plant, although the nature of their interactions is unknown in most cases [22,23]. Some of these microorganisms are even considered as natural biocontrol agents due to their ability to protect the plant against phytopathogens and reinforce natural plant defenses [24].

In previous studies, a great diversity of endophytes belonging to different taxa has been documented on grapevine plants [25,26]. In this line, in viticulture and oenology the complex of the microbiota present is recognized by a major imprint on the regional local “terroir” [27]. Moreover, the endophytic community, particularly yeast and bacterial species carried within the plant, plays a role in early fermentation stages and is linked to geographical origin—contributing to the distinct characteristics of wines from different regions [28]. In this study, we explored the culturable mycobiota associated with *Xff* under field conditions. The analysis was limited by significant variation in bleeding time among plants and by the low volume of sap collected, which, in most cases, was insufficient for further study. Nonetheless, different fungi, including yeast-like organisms, were successfully isolated from the sap of five plants—three of which tested positive for *Xff*. The recovered microorganisms included both beneficial and potentially pathogenic fungi. Notably, some beneficial fungi may be of agronomic interest in integrated pest management (IPM). For example, *Rhodotorula mucilaginosa* and *Aureobasidium pullulans*, both isolated in this study, have been previously reported as promising biocontrol agents against grapevine trunk diseases (GTDs) [29]. Furthermore, *A. pullulans* has shown efficacy in managing bitter rot in grapes [30]. Yeast and yeast-like microorganisms have also been investigated as biological control agents, particularly for their role in shaping indigenous yeast populations during spontaneous fermentation [29].

Among the more concerning fungal isolates were those associated with GTDs, which are now recognized as a major biotic threat to grapevine health. The necrotrophic pathogen *Sclerotinia sclerotiorum (Ss)*, which causes shoot rot in grapevines, was identified and is typically active during spring, when mild temperatures prevail [20]. Other genera recovered in this study—such as *Alternaria*, *Phoma*, *Cladosporium*, and the *Diaporthe/Phomopsis* complex—are also capable of becoming pathogenic under favorable conditions. Studies have shown that species of *Alternaria* can cause berry rot, mold, and pedicel and rachis diseases [31], with *A. alternata* identified as a postharvest pathogen and a cause of leaf spot [32]. Members of the *Phoma* complex have previously been implicated in the decline and death of young grapevines [33,34].

According to the preliminary analysis of fungal identity analyzed here, it is highlighted that some fungal taxa belonging to the Ascomycetes group overlap at the genus and species level due to their similar morphology, often producing small, spherical or ellipsoid conidia (with or without septa) and pycnidial conidiomata, and forming dark, felted colonies. Due to these morphological similarities, accurate species-level identification requires molecular confirmation. Recent studies [35] compared the biodiversity of fungi in *Vitis vinifera* by both traditional and molecular methods to obtain a better resolution in species identification, richness and distribution. These researchers concluded that a combination of both approaches (i.e., traditional and culture-independent) is needed for proper evaluation. In agreement, in our work, the traditional technique using morphobiometrical analysis of colonies and conidia was useful as a preliminary exploratory approach to assess fungal diversity. However, for accurate identification of the recovered taxa, complementary molecular techniques will be necessary.

Differences in the fungal communities of *Xff*-positive and *Xff*-negative plants were observed, with community richness varying according to the presence of the bacterium. These findings could contribute to the understanding of the roles played by each fungal group in ecosystem stability and function.

Some of the taxa identified in this study—such as the *Botryosphaeria* complex (Botryosphaeriales), *Phaeoacremonium/Phaeomoniella* spp. (Togniniales), and dark-spored fungi with pycnidial conidiomata (e.g., *Phoma*, *Phomopsis/Diaporthe* complex)—are consistent with GTD-associated pathogens. These fungi invade the xylem, block sap flow (tracheomycosis or hadromycosis), and lead to plant decline. The symptoms they cause—such as reduced productivity, arm dieback, progressive deterioration, graft failure, shoot die-off, chlorosis, necrosis, white rot, and wedge-shaped trunk discoloration—overlap with those of *Xff* infection [36].

Little is known about how *Xff* interacts with other components of the plant microbiome during co-infections. Few studies have explored the synergic effects between pathogenic microorganisms in plant disease development under co-infections [37,38]. For example, previous work by Araujo et al. (2002) [9] reported that the presence of the endophyte *Methylobacterium* had a synergistic effect causing an increase in the occurrence and intensity of symptoms induced by *Xff* subsp. *pauca,* in *Citrus sinensis*. In this work, we focused on the interaction between a fungal pathogen found in the xylem sap, *Ss*, which was recently registered on grapevines in Mallorca causing shoot blight, and *Xff* [20]. The infection of grapevines by *Xff* yielded plants more vulnerable to *Ss*, which could have been the result of a detrimental effect of *Xff* on the plant’s defense systems. Although *Xff* lacks a Type III secretion system, that suppresses the host plant defense responses [39], we hypothesize that the increased virulence of *Ss* in *Xff*-infected grapevines could rather be due to a debilitated plant’s metabolism by obstructing xylem vessels, thereby reducing stomatal function, carbon assimilation and plant growth [40].

In terms of physiological parameters, Ss appeared to override the effects of *Xff* infection. Both pathogens influenced stomatal movement in an antagonistic manner. *Xff* has been reported to cause stomatal closure, possibly due to its colonization of xylem vessels and the plant’s production of tyloses to contain the infection. This leads to reduced hydraulic conductivity and water stress [40]. Additionally, *Xff* may trigger abscisic acid (ABA) signaling, which contributes to stomatal closure and may suppress plant defenses [41]. Such closure also limits photosynthesis and transpiration, preventing evaporative cooling and raising leaf temperature [42,43]. Conversely, according to Guimarañes and Stotz [44], Ss may induce stomatal opening via oxalic acid secretion. This disrupts guard cell function and ABA-mediated closure, potentially resulting in foliar dehydration. This mechanism could explain why the stomatal-opening effect of Ss predominated over the closure induced by *Xff* infection, effectively counteracting the impact of ABA.

To conclude, this preliminary screening of the grapevine sap endophytes under *Xff* infection, and the symptoms observed in concomitant infection *S. sclerotiorum*, underscores the importance of taking into consideration the microbiota–pathogen interaction when studying plant diseases under field conditions. The results here also alert us to the underdiagnosed and underestimated components of microbiota, especially fungi co-isolated with bacteria, which are frequently dismissed as irrelevant despite their potential to influence disease outcomes.

## 4. Materials and Methods

### 4.1. Experimental Design

To study the biodiversity of sap grapevine fungal endophytes in *Xff*-affected vineyards, a survey was conducted in a commercial vineyard located in a typical wine production region in the island of Mallorca (Spain) in 2020. Selection of grapevines was performed based on the analysis of a total of 50 grapevines cv. Cabernet Sauvignon analyzed from that place in summer in 2019 to identify *Xff*-positive plants, according to protocols shown in Section 4.2.

### 4.2. PCR Assays to Test the Presence of Xff

DNA extraction from plant extracts was performed from leaf veins and petioles, and xylem sap using an EZNA HP Plant Mini kit (Omega-Biotek, Norcross, GA, USA) following the manufacturer’s instructions, as described in the EPPO protocol (EPPO, 2016) [45]. The presence of Xff was assessed by real-time PCR using two specific protocols with primers XF-F/XF-R and the TaqMan probe XF-P (Harper et al., 2010) [46] and primers HL5/HL6 and the TaqMan probe HL-P (Francis et al., 2008) [21] using an Applied Biosystems QuantStudio 3 Real-Time PCR System (Thermo Fisher Scientific, Waltham, MA, USA) [21,46]. Primers and qPCR conditions were as follows: For Harper qPCR, the target sequence is located in the gene coding for the 16S rRNA processing RimR protein. Sequences were as follows: forward primer, XF-F, 5′-CACGGCTGGTAACGGAAGA-3′; reverse primer, XF-R, 5′-GGGTTGCGTGGTGAAATCAAG-3′; and sequence of probe, XF-P, 5′-6-FAM-TCG CATCCCGTGGCTCAGTCC-BHQ-1-3′. PCR conditions: pre-incubation of 50 °C for 2 min, an initial denaturation of 95 °C for 10 min, followed by 40 cycles of 95 °C for 10 s and 62 °C for 40 s. Heating ramp speed: 1.6 °C/s. For Francis qPCR, the target sequence is a conserved hypothetical protein HL gene. The sequence primers are as follows: forward primer HL5: 5′-AAGGCAATAAACGCGCACTA-3′; reverse primer HL6: 5′-GGTTTTGCTGACTGGCAACA-3′; the probe sequence is 5′-6-FAM-TGGCAGGCAGCAACGATACGGCT-BHQ-1-3′. Pre-incubation (UNG step) at 50 °C for 2 min, initial denaturation at 95 °C for 10 min, followed by 45 cycles of 95 °C for 15 s and 60 °C for 60 s. Heating ramp speed: 1.6 °C/s. All samples were analyzed in triplicate, including the positive and negatives controls.

### 4.3. Sap Collection

The sap samples were collected in the vineyard during grapevine bleeding. Ten plants of Cabernet Sauvignon were selected; nonetheless, sufficient xylem sap to continue the study was obtained from five plants only. To collect the bleeding sap, the remaining canes left after pruning were cut off at 2–3 cm, and a few mL of sap were allowed to drop to clean the cut before attaching a collection tube to the cut end. Sap samples were kept at 4 °C until use. Sap aliquots were used to confirm the presence of *Xff*, according to protocols shown on the previous Section. One hundred µL of each sap sample were plated on Sabouraud agar and incubated for 7 days at 30 °C.

### 4.4. Fungal Identification

The microorganisms were identified using conventional methods that involved isolating and cultivating them on artificial media and then classifying them according to their taxonomy. Cultivable fungi were isolated and incubated in Petri dishes with PDA 2% medium or Sabouraud medium for 3–7 days in a growth chamber at 25 °C, 12 h of light and 12 h of darkness.

Each of the different morphologically identified colonies was transferred through a small agar disk (about 5 mm2) of the growing fungus to a fresh 60 mm diameter PDA plate. The obtained colonies were grouped and numbered according to their morphological characteristics, based on shape, form, size, growth time, border, surface, opacity, pigmentation, and the shape and size of the fungal fruiting bodies, spores, and hyphae. Additionally, shoot tissues and material collected with visible symptoms were placed in a moist chamber, and direct observation of leaf symptoms were carried out.

With the use of specific keys, microorganisms derived from colonies were identified by microscopic examination of mycelia and spores, morphobiometric traits, and cultural features. A group-level taxonomy classification was carried out, which considered the identification of endophytes, which are beneficial microorganisms, and risk genera linked to pathogenic fungi on wood trees that have not been previously reported on grapevines in Mallorca, as well as genera that are still poorly known despite their importance as plant pathogens. Representative cultures were deposited at the UIB culture collection. The current name of the microorganism was used according to Index Fungorum from 2018 [47].

### 4.5. Co-Inoculation Assays to Test the Interaction Xylella fastidiosa subsp. fastidiosa (Xff) with Sclerotinia sclerotiorum (Ss)

The manifestation of symptoms in plants infected with *Xff* interacting with the fungal pathogen Ss, recently identified as a new potential biotic adversity for grapevines at Mallorca [20], was examined.

Two-year-old potted grapevine plants cv. Cabernet Sauvignon were used. A year earlier, half of the plants had been inoculated with the strain of *X. fastidiosa* subsp. *fastidiosa* RTA821. The inoculum was prepared from 10-day colonies grown on BCYE agar. A milky solution was prepared using Ringer’s solution to achieve a suspension of approximately 108 cell mL^−1^, which was immediately used, as it tended to precipitate. Inoculations were performed at the lowest node of each branch, where a 10 µL droplet was applied using a needle inoculation technique by pin pricking until complete absorption of the drop was observed. The other half of plants were mock inoculated using 10 µL of Ringer’s solution.

A completely randomized design was employed, consisting of four treatments with four replicates per treatment: (1) non-infected plants (Mock) (2), plants previously infected with *Xff*, (3) plants inoculated with *Ss*, (4) plants previously infected with *Xff* and inoculated with *Ss*.

*Sclerotinia* artificial inoculations were carried out in 8 points of 2 branches/plant and treatment for fungal infection. The inoculum consisted of placing agar discs with actively growing fungal mycelium on fresh plant wounds and then covering this inoculation zone with parafilm. Each inoculated branch was covered with nylon bags for 48 h to prevent desiccation and increase humidity. The successful infection of *Xff* was confirmed by PCR technique as described above (4.2). Fungal colonization was quantified by the length and number of branches with rot symptoms and the number of petioles showing rot symptoms at 7 days post infection (dpi). Regarding morphological measurements of disease evolution, the number of lesions on the stems, the length of necrotic lesions and the number of necrotic petioles, were measured in each plant, twice per week. The AUDPC was calculated according to Madden et al. (2007) [48].

### 4.6. Chlorophyll SPAD and Stomatal Conductance

Plant disease progression was assessed by physiological and morphological measurements. At physiological level, leaf total chlorophyll concentration using a portable chlorophyll meter (SPAD Model CL-01, Hansatech Instruments, King’s Lynn, Norfolk, UK) and stomatal conductance (gs) measured using a Leaf Porometer (Model SC-1Decagon Devices, Inc., Pullman, WA, USA) were measured on leaves located above and below the *Ss* inoculation point. Measurements were performed once a week, from 10 to 13 h.

### 4.7. Statistical Analysis

The statistical software InfoStat 2020 was used for data analysis [49] for the one-way ANOVA and Tukey’s test (*p* < 0.05). For the interaction assay, a CPA analysis was performed using XLSTAT software 2021.

## Figures and Tables

**Figure 1 plants-14-01976-f001:**
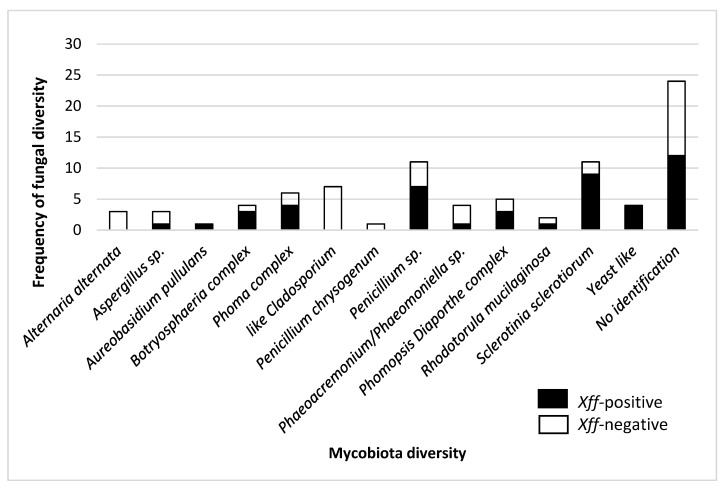
Mycobiota diversity in xylem sap collected from a vineyard in Mallorca, Spain. Data from 86 fungi isolates, obtained from two plants that tested positive and three plants that tested negative for the occurrence of *Xff.* No identification: fungal genus unclassified and referred to as unidentified taxon.

**Figure 2 plants-14-01976-f002:**
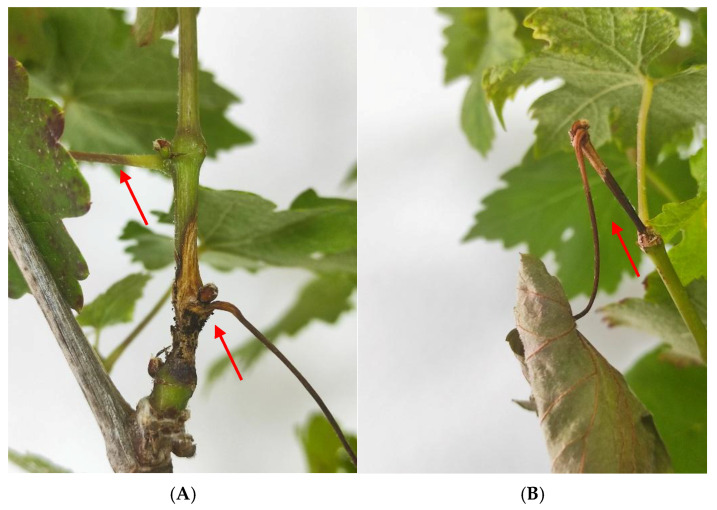
Symptoms of *S. sclerotiorum* in Cabernet Sauvignon grapevines at 7 dpi. (**A**) Broken stems from the point of infection because of the injury. Necrotic lesions (arrows), brown in color, with darker edges extended along the stem that remained green, compromising petioles that were totally or partially necrotic and therefore led to the wilting of the leaves in the affected shoot (**B**) detail of necrotic petiole and leaf wilting.

**Figure 3 plants-14-01976-f003:**
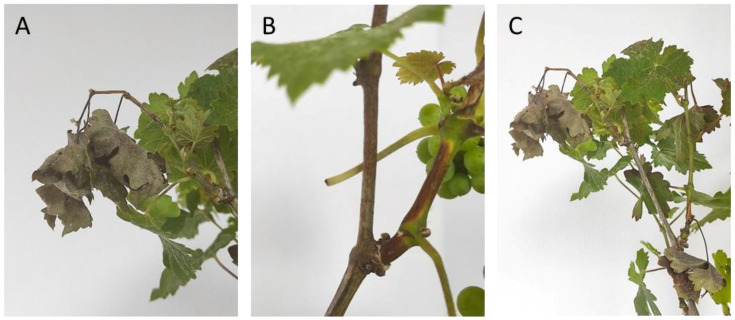
Symptoms of Ss in C. Sauvignon grapevine at 7 dpi (**A**) Epinasty and necrosis in the stem with total or partial decay of the shoot or its leaves. (**B**) Detail of the necrotic lesion in a young shoot fractured due to spontaneous breaking. (**C**) Epinasty and distal wilting in terminal shoot’s leaves.

**Figure 4 plants-14-01976-f004:**
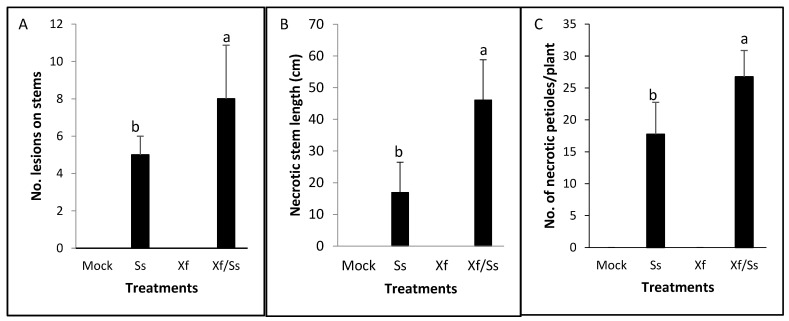
Symptoms on grapevine plants control (Mock), infected with *Sclerotinia* (*Ss*), *Xylella* (*Xff*) and the combination (*Ss + Xff*). (**A**) Number of necrotic lesions on stems; (**B**) necrotic stem length and (**C**) number of necrotic petioles per grapevine plant analyzed. Measurements of symptoms were performed at 7 dpi. The bars represent the average of each measure. The experiment followed a completely randomized design with four replicates per treatment. Statistical analysis using one-way ANOVA revealed a significant effect: In (**A**): F = 5.65, *p* = 0.0119, df = 15; (**B**): F = 29.67, *p* ≤ 0.0001, df = 15; (**C**): F = 68.28, *p* < 0.0001, df = 15. Different letters indicate statistically significant differences (*p* < 0.05).

**Figure 5 plants-14-01976-f005:**
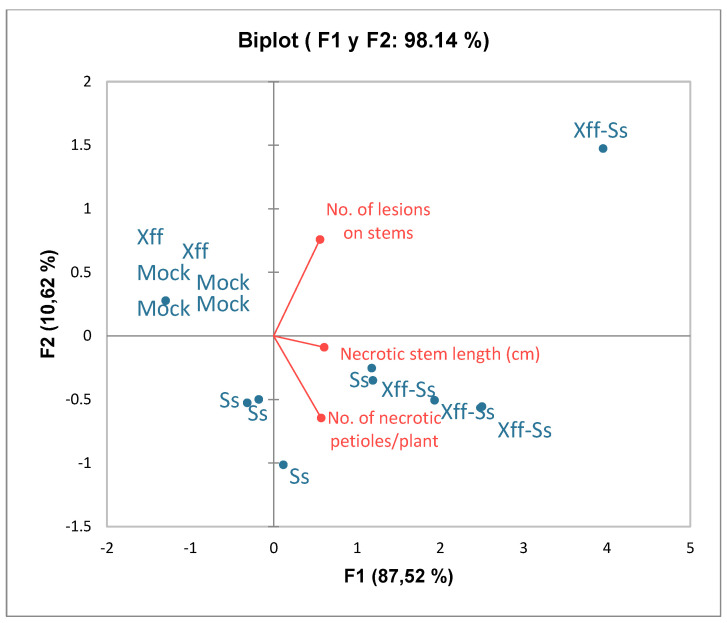
Principal component analysis (PCA) biplot showing the distribution of treatments and disease severity variables in infected and mock-inoculated plants. Plants control (Mock), infected with *S. sclerotiorum* (*Ss*), *X. fastidiosa* (*Xff*) and the combination (*Ss + Xff*).

**Figure 6 plants-14-01976-f006:**
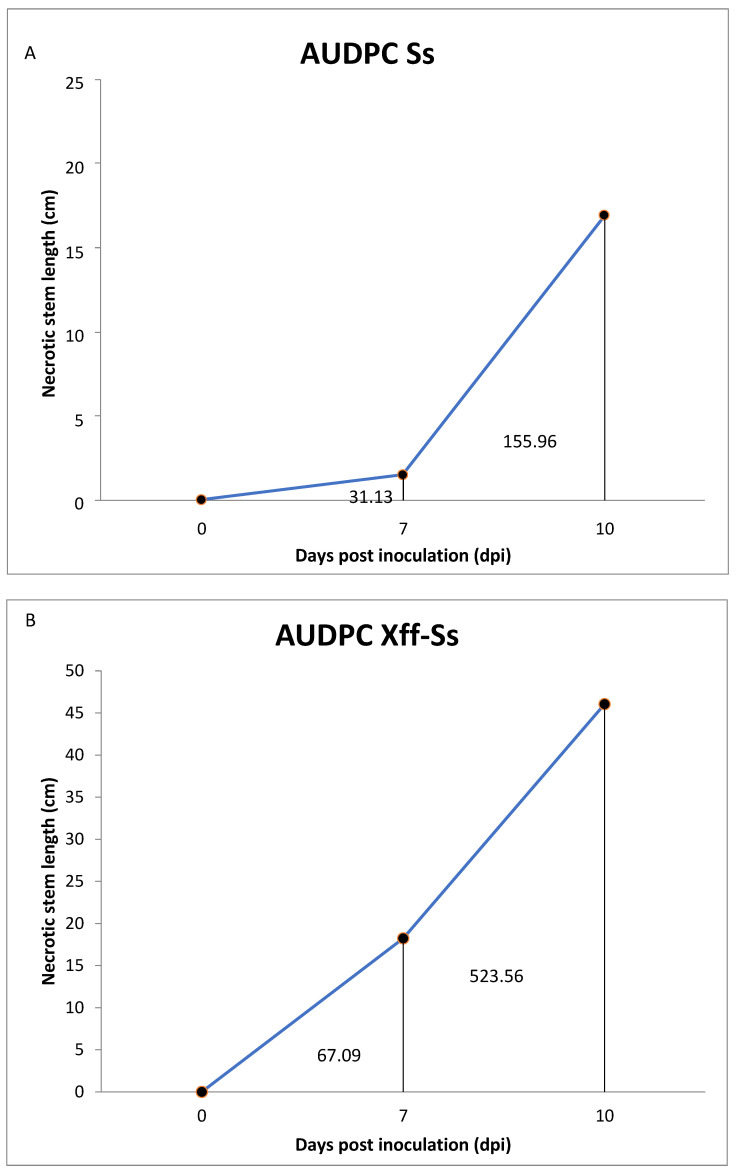
Area Under the Disease Progress Curve (AUDPC) for necrotic stem length in two treatments. Disease progression was assessed at 0-, 7-, and 10-days post inoculation (dpi) and quantified as necrotic stem length (cm). Panel (**A**) represents the AUDPC for plants inoculated with Ss, and Panel (**B**) corresponds to plants inoculated with *Xylella fastidiosa* subsp. *fastidiosa* + *Ss* (*Xff-Ss*). AUDPC values were calculated using data from four biological replicates per treatment. The *Xff-Ss* treatment exhibited a significantly higher disease severity, with an average AUDPC value of 523.56 ± 278.64, compared to 155.96 ± 83.31 in the Ss treatment (mean ± standard error). Statistical analysis using one-way ANOVA revealed a significant difference between treatments (F = 11.54, *p* = 0.0008, df = 15). These results suggest a synergistic or enhancing effect of *Xff* on disease development when co-inoculated with Ss.

**Figure 7 plants-14-01976-f007:**
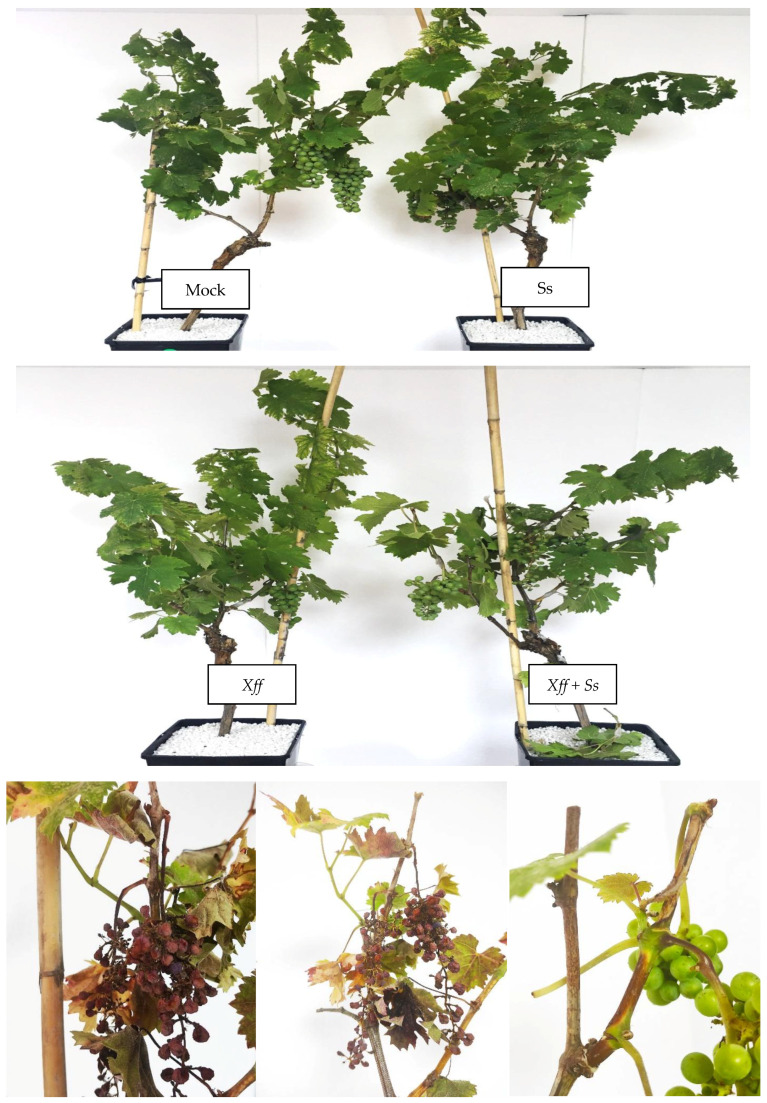
Symptoms on Cabernet Sauvignon grapevine plants at 7 dpi in Mock, *Ss*, *Xff* and *Ss + Xff*. Details of grape bunches and stem necrosis, stem breakage, wilting and falling of leaf petioles caused by *Ss* in plants inoculated with *Xff* (below).

**Figure 8 plants-14-01976-f008:**
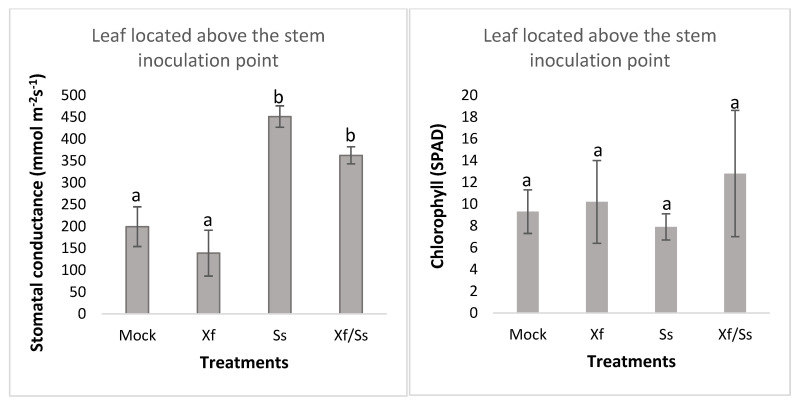
Stomatal conductance and chlorophyll content in grapevine leaves across different treatments. Physiological parameters were measured at 10 days post inoculation in grapevine plants subjected to four treatments: Control (Mock), *Xylella fastidiosa* subsp. *fastidiosa* (*Xff*), *Sclerotinia sclerotiorum* (*Ss*), and co-inoculation with *X. fastidiosa* + *S. sclerotiorum* (*Xff*/*Ss*). Chlorophyll content (SPAD units) differed significantly among treatments (F = 2.01, *p* = 0.1664, df = 15). The highest mean value was observed in *Xff*/*Ss* (12.82 ± 4.69), followed by *Xff* (10.19 ± 3.01), Mock (9.26 ± 1.63), and *Ss* (7.73 ± 1.30). Tukey’s test indicated that *Xff*/*Ss* was significantly different from *Ss* (*p* < 0.05). Stomatal conductance (mmol m⁻² s⁻¹) also varied significantly among treatments (F = 90.69, *p* < 0.0001, df = 15). The Ss treatment exhibited the highest conductance (451.1 ± 20.1), significantly higher than *Xff*/*Ss* (362.3 ± 8.1), Mock (199.4 ± 36.3), and *Xff* (139.0 ± 42.7), with all pairwise differences significant (*p* < 0.05). Bars represent the mean ± standard deviation of four biological replicates per treatment. Different letters indicate statistically significant differences according to Tukey’s HSD test (*p* < 0.05).

## Data Availability

Data are contained within the article and Appendix A.

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
