# Peer review of "Xylem Sap Mycobiota in Grapevine Naturally Infected with Xylella fastidiosa: A Case Study: Interaction of Xylella fastidiosa with Sclerotinia sclerotiorum"

_plants, 2025, doi:10.3390/plants14131976_

Round 1
Reviewer 1 Report
Comments and Suggestions for Authors
This manuscript explores an interesting and timely topic: the interaction between Xylella fastidiosa (Xff), a major xylem-limited plant pathogen, and fungal components of the grapevine sap mycobiome, with a particular focus on Sclerotinia sclerotiorum (Ss). The authors aim to shed light on whether co-occurrence of these pathogens in the xylem could lead to synergistic effects that exacerbate grapevine disease. This topic is particularly relevant for viticulture in Mediterranean regions, where Xff is an emerging threat. Overall, I find the research question to be original and worth pursuing. The idea of studying the fungal microbiome in plants naturally infected by Xff and experimentally assessing its interaction with a known necrotrophic fungus is scientifically sound. However, the execution of the study presents several critical limitations that need to be addressed to ensure the findings are robust and meaningful.
One of the main concerns is the use of exclusively culture-based techniques to characterize the fungal community. This approach greatly limits the scope and accuracy of fungal identification, especially when dealing with complex genera such as Phoma, Cladosporium, or Alternaria, which often contain cryptic species. Incorporating molecular methods such as ITS sequencing—or at the very least sequencing representative isolates—would significantly strengthen the taxonomic reliability of the results.
The sample size is very small, which limits the statistical power and the general applicability of the findings. While this is acknowledged by the authors, it remains a significant limitation, especially for a study that attempts to draw ecological and pathophysiological conclusions.
The co-inoculation experiment involving Xylella and Sclerotinia is conceptually strong but needs clearer methodological details. For example, how was successful Xff infection confirmed post-inoculation? Were the plants randomized? Were other microbial contaminants excluded? How was the fungal colonization quantified or observed beyond symptom assessment? Including such information would help readers better interpret the experimental results.
The authors suggest a synergistic effect between Xff and Ss based on increased disease severity and altered stomatal conductance. This is plausible and intriguing, but it remains speculative without supporting evidence such as histological observations, quantification of pathogen load, or analysis of plant defense responses. Some discussion points in the manuscript go beyond the data and would benefit from a more cautious and evidence-based interpretation.
The manuscript requires substantial editing for grammar, clarity, and style. There are numerous typographical and grammatical issues that hinder readability. The abstract, in particular, is too long and convoluted, it should be rewritten to focus on key objectives, methods, and findings in a concise way.
Some figures lack proper labeling, and axis titles are missing. Statistical results (e.g., ANOVA, PCA) are mentioned but not fully explained in the figure legends. Including detailed statistical output (F-values, p-values, degrees of freedom) would improve transparency and replicability.
Author Response
Reviewer 1
Comment 1: One of the main concerns is the use of exclusively culture-based techniques to characterize the fungal community. This approach greatly limits the scope and accuracy of fungal identification, especially when dealing with complex genera such as Phoma, Cladosporium, or Alternaria, which often contain cryptic species. Incorporating molecular methods such as ITS sequencing—or at the very least sequencing representative isolates—would significantly strengthen the taxonomic reliability of the results.
Reply 1: Although molecular techniques such as ITS sequencing could offer more precise taxonomic resolution, the present study opted for morpho-cultural methods as a valid alternative based on the objectives and available resources, as culture-based methods are more cost effective and do not need specialized equipment. Moreover, morpho culture-based techniques to characterize a fungal community is a methodology broadly accepted nowadays, successfully used previously by other studies, especially in a preliminary step of microbial prospection like here. In future research we would like to include molecular techniques for taxonomical identification of some isolated beneficial microorganisms under our interest in a deeper study of biological control.
Also, due to limitations in funding and access to high-throughput sequencing platforms, molecular identification was not feasible within the scope of this work.
So, as the primary aim of this study was to assess the culturable mycobiota associated with grapevine under Xylella stress, culture-based techniques of microorganisms remain appropriate and informative.
Comment 2: The sample size is very small, which limits the statistical power and the general applicability of the findings. While this is acknowledged by the authors, it remains a significant limitation, especially for a study that attempts to draw ecological and pathophysiological conclusions.
Reply 2: While the small sample size limits statistical robustness and the ability to capture the full spectrum of grapevine microbial diversity, the results remain informative from an ecological and phytopathological point of view. The findings here are consistent and offer meaningful insights into microbioma dynamics under Xylella stress. These initial data help lay the foundation for future work and emphasize the importance of integrating microbial ecology into plant disease frameworks.
Our limitation arises primarily from logistical challenges inherent to field-based plant pathology studies, including to have enough grapevines bleeding during the sampling time a sufficient sap volume to conduct the study. When grapevine bleeding season ended, we had no other option but to use the data collected from one cultivar in one location, the one that had a low, but still sound number of Xylella fastidiosa positive and negative samples. Despite these challenges, the study provides valuable preliminary insights that can inform future research.
Importantly, the aim of this work was exploratory in nature, intended to identify potential patterns between plants with positive or negative Xylella infection in interaction with beneficial or pathogenic fungus, and generate hypotheses rather than to establish definitive causal relationships or make broad generalizations, especially when working with understudied systems or emerging phenomena like Ss on grapevine in Mallorca.
Moreover, our desire is to use this preliminary work to draw the attention of financial institutions towards this problem, commonly present in wine production regions, to get financial support to conduct a thorough study following this line of research.
Comment 3: The co-inoculation experiment involving Xylella and Sclerotinia is conceptually strong but needs clearer methodological details. For example, how was successful Xff infection confirmed post-inoculation? Were the plants randomized? Were other microbial contaminants excluded? How was the fungal colonization quantified or observed beyond symptom assessment? Including such information would help readers better interpret the experimental results.
Reply 3: Plants were first prick-pin inoculated with Xff as was described in our first version: “A year earlier, half of the plants had been inoculated with the strain of X. fastidiosa subsp. fastidiosa RTA821”.
The successful infection was confirmed by PCR. Plants in experiments were randomized (these details were added in the manuscript: “A completely randomized design was employed, consisting of four treatments with four replicates per treatment”). No microbial contaminants were observed before or after the inoculation of the target fungus Ss due to experiments involving Xff require strict containment and biosafety procedures due to the pathogen’s quarantine status, the experiments were carried out in a laboratory Biosafety Level 3(BSL-3) and used healthy host plants. The colonization of the fungus on grapevine plants was quantified by three variables: number of lesions on stems, necrotic stem length (cm) and number of necrotic petioles/plants, like is shown in Fig 4 and Fig 5. Moreover, the fungal colonization was quantified and observed beyond symptom assessment progress by AUDPC (Fig. 6).
Figure 4. Symptoms on grapevine plants control (Mock), infected with Sclerotinia (Ss), Xylella (Xff) and the combination (Ss+Xff). (A) Number of necrotic lesions on stems; (B) necrotic stem length and (C) number of necrotic petioles per grapevine plant analyzed. Measurements of symptoms were performed at 7 dpi. Different letters Indicate statistically significant differences (p<0.05).
Figure 5. PCA biplot graph representing the three variables measured (number of necrotic petioles, number of lesions on stems, and total length of the lesions) in the interaction Xff-Ss on grapevine Cv. Cabernet Sauvignon.
Figure 6. AUDPC. Graphic representation of the total accumulated disease measured as the area under disease progress curve in both treatments: Xff-Ss (A) and Ss (B). Data obtained from three evaluation diseases time points.
Comment 4: The authors suggest a synergistic effect between Xff and Ss based on increased disease severity and altered stomatal conductance. This is plausible and intriguing, but it remains speculative without supporting evidence such as histological observations, quantification of pathogen load, or analysis of plant defense responses. Some discussion points in the manuscript go beyond the data and would benefit from a more cautious and evidence-based interpretation.
Reply 4: The synergistic effect between Xff and SS suggested in the text is supported by statistically significant evidence of increased disease severity and the altered stomatal conductance when both microorganisms are co-inoculated. The authors consider these two observations bring enough evidences to maintain this interpretation. Moreover, our suggestion is in accordance with reviewer 3 that said “The paper convincingly demonstrates that the composition of the microbiome can both support or exacerbate disease progression in X. fastidiosa-infected plants”.
Finally, notwithstanding would be a complement, was not objective of this paper to elucidate the underlying mechanisms (e.g plant defense responses, histological observations).
Comment 5: The manuscript requires substantial editing for grammar, clarity, and style. There are numerous typographical and grammatical issues that hinder readability. The abstract, in particular, is too long and convoluted, it should be rewritten to focus on key objectives, methods, and findings in a concise way.
Reply 5: grammar, clarity and style were considered and improved in the new submission.
The abstract was rewritten focusing on objectives, methods and findings in a more concise way.
Grapevine (Vitis vinifera) is a key crop in Mediterranean agriculture, now increasingly threatened by Xylella fastidiosa subsp. fastidiosa (Xff), the causal agent of Pierce’s disease. This study investigated: (1) the diversity of culturable fungal endophytes in the xylem sap of naturally Xff-infected grapevines, and (2) the interaction between Xff and the pathogenic fungus Sclerotinia sclerotiorum identified in the sap.The xylem sap was collected from Cabernet Sauvignon vines in Mallorca, Spain, and fungal communities were characterized using culture-dependent methods. Both beneficial fungi (e.g., Aureobasidium pullulans, Rhodotorula mucilaginosa) and pathogenic species (e.g., S. sclerotiorum, Cladosporium, Alternaria alternata, and the Phoma complex) were isolated from both Xff-positive and Xff-negative plants, indicating similar community profiles. Although limited by small sample size, these findings offer preliminary evidence of complex ecological interactions between Xff and the xylem-associated mycobiota, with potential implications for grapevine health and disease development under varying environmental and management conditions. Further experiments under controlled conditions revealed that grapevines co-inoculated with Xff and S. sclerotiorum showed increased disease severity, suggesting a synergistic interaction. These preliminary results highlight the complex interplay between Xff and the fungal endophytic microbiome, which may modulate grapevine susceptibility depending on environmental and management conditions.
Comment 6: Some figures lack proper labeling, and axis titles are missing. Statistical results (e.g., ANOVA, PCA) are mentioned but not fully explained in the figure legends. Including detailed statistical output (F-values, p-values, degrees of freedom) would improve transparency and replicability.
Reply 6: The labeling, axis and titles were re checked and mistakes corrected in figures 1 and 8. Detailed statistical output were added in all the graphics.

Reviewer 2 Report
Comments and Suggestions for Authors
Dear Authors,
Manuscript ID: plants-3659204. The manuscript as an Article entitled ‘Xylem sap mycobiome in grapevine naturally infected with Xylella fastidiosa. A case study: Interaction of Xylella fastidiosa with Sclerotinia sclerotiorum’ was submitted by Analía Perelló et al. to Plants.
Researchers are analyzing grapevine infected with Xylella fastidiosa subsp. fastidiosa (Xff), a bacterium responsible for serious plant diseases and Xff interaction with endophytic just culturable xylem sap fungus. It is very interesting and potentially useful research. When studying only culturable mycobiota, however molecular identification or metagenomic methods were not used for more accurate identification of mycobiota, but this is discussed in the Discussion section. Although it is good that molecular methods were used for Xylella identification. Note. You should write in the results section, perhaps in separate section 2.1, that you identified xylella by molecular methods, i.e. PCR. Or for example in line 120 and 125 'tested positive or negative for the occurrence of Xff', write that tested by PCRs as described in the methods. Thus, not only typing corrections but Major revisions are needed. Below are some of the necessary minor corrections.
Title: Xylem Sap Mycobiome? Same is in Abstract: Line 28, 31 etc. As you did analysis of culturable fungal endophytic communities, you should use term ‘mycobiota’ instead ‘microbiome or mycobiome. When you are analyzing fungus together with Xff bacterium than use ‘microbiota’. You did not do metagenomic next generation sequence (NGS) analysis, did it? Correct the Title and in the text, please.
Line 90. mycobiome of the xylem sap of the vineyard sampled in this current study – mycobiota.
Line 126, the structure of the microbiome recovered – microbiota.
Line 144, Figure 1. Mycobiome diversity – mycobiota.
Line 249, three of then identified as – three of them.
Line 250, recoveredincluded – two separate words.
Line 350. I am reminding you once more. You wrote in the Methods: 4.2. PCR Assays to Test the Presence of Xf. But I cannot find that you were using PCR in the Results. Write It in the results if it is missing, please.
After these corrections the manuscript could be published in Plants’ journal.
Sincerely, May 19, 2025
Author Response
Reviewer 2
Major corrections
Comment 1: Molecular identification or metagenomic methods were not used for more accurate identification of mycobiota, but this is discussed in the Discussion section. Although it is good that molecular methods were used for Xylella identification.
Reply 1: Although molecular or metagenomic techniques can provide a more detailed resolution of fungal community composition, this study employed traditional morphological and/or culture-based methods due to practical, economic, and methodological constraints. These approaches remain widely used and provide sufficient taxonomic resolution for the study's objectives, particularly when focusing on culturable or ecologically dominant species from grapevine infected with Xylella. Moreover, although full metagenomic analysis was not performed here, representative isolates of interest in biological control would be identified using ITS secuencing in future studies to support morphological identification and to complement the current findings.
Comment 2: You should write in the results section, perhaps in separate section 2.1, that you identified xylella by molecular methods, i.e. PCR. Or for example in line 120 and 125 'tested positive or negative for the occurrence of Xff', write that tested by PCRs as described in the methods.
Reply 2: In the results section (2.1) we added a paragraph with the molecular identification of Xylella
Minor corrections:
Comment 3: Title: Xylem Sap Mycobiome? Same is in Abstract: Line 28, 31 etc. As you did analysis of culturable fungal endophytic communities, you should use term ‘mycobiota’ instead ‘microbiome or mycobiome.
Reply 3: the word mycobiome was changes by mycobiota, as suggested.
Comment 4: When you are analyzing fungus together with Xff bacterium than use ‘microbiota’. You did not do metagenomic next generation sequence (NGS) analysis, did it? Correct the Title and in the text, please.
Reply 4: the title and text were corrected.
Comment 5: Line 90. mycobiome of the xylem sap of the vineyard sampled in this current study – mycobiota.
Reply 5: mycobiome was changed by mycobiota
Comment 6: Line 126, the structure of the microbiome recovered – microbiota.
Reply 6: microbiome was changed by microbiota.
Comment 7: Line 144, Figure 1. Mycobiome diversity – mycobiota.
Replay 7: mycobiome was changed by mycobiota in legend and figure axis
Comment 8: Line 249, three of then identified as – three of them.
Replay 8: the mistake was corrected (then by them)
Comment 9: Line 250, recoveredincluded – two separate words.
Replay 9: the two words were separated.
Comment 10: Line 350. I am reminding you once more. You wrote in the Methods: 4.2. PCR Assays to Test the Presence of Xf. But I cannot find that you were using PCR in the Results. Write It in the results if it is missing, please.
Replay 10: we agree and the PCR results were highlighted in the results section point 2.1, according with this suggestion.

Reviewer 3 Report
Comments and Suggestions for Authors
The paper addresses a highly relevant and timely topic concerning the interactions between the plant microbiome and pathogens, with a particular focus on Xylella fastidiosa, which is classified as a quarantine organism in the European Union. The authors adopted an interesting research approach by combining field analyses (isolation of xylem endophytes) with a controlled greenhouse experiment involving co-infection of grapevines with X. fastidiosa and Sclerotinia sclerotiorum. An additional strength of the study is the inclusion of physiological measurements, such as stomatal conductance and chlorophyll content (SPAD).
The study enabled the identification of potential antagonists of grapevine trunk diseases (GTDs), such as Aureobasidium pullulans, as well as the identification of possible risk factors associated with co-infections. The paper convincingly demonstrates that the composition of the microbiome can both support or exacerbate disease progression in X. fastidiosa-infected plants.
Despite its scientific value, the manuscript lacks clarity in some sections – particularly in the Introduction and Discussion – which makes it difficult to follow the argumentation. Improving the structure and avoiding repetition of information already presented in the Abstract and Results is recommended.
There are also some errors in the taxonomy section that should be corrected. For example, the Botryosphaeria complex was incorrectly assigned to the order Hypocreales, whereas it belongs to Botryosphaeriales (class Dothideomycetes). The term "Dematiaceous Coelomycetes" is outdated; it is recommended to use more current terminology, such as "pycnidial dark-spored fungi", or to refer to modern taxonomic groups (e.g., Phoma spp., Diaporthe spp.). Similarly, the use of "Deuteromycetes" in the Results section should be clarified, as it is no longer recognized as a valid taxonomic category.
In the section on physiological results, it would be helpful to indicate whether the differences in stomatal conductance were statistically significant, and to specify the number of plants per treatment group (n).
The manuscript also requires thorough language editing, both grammatically and stylistically. Many sentences are overly long or unnecessarily complex. There are also redundant statements – for instance, the role of Aureobasidium and Rhodotorula is described in similar terms several times. Care should be taken to ensure clarity in data presentation – for example, the notation "5:8 vs 8:8" needs to be clearly explained.
Finally, the methodological and statistical sections should be improved. The manuscript lacks details on the number of replicates used in the greenhouse experiment, the p-values from statistical tests, and whether the assumptions of ANOVA were met. While the PCA was appropriately applied, it would benefit from a more detailed description of its parameters and interpretation.
Comments on the Quality of English Language
The manuscript is understandable, but would benefit from language editing to improve clarity and flow. Minor grammatical and stylistic revisions are recommended to enhance overall readability.
Author Response
Reviewer 3
Comment 1: Despite its scientific value, the manuscript lacks clarity in some sections – particularly in the Introduction and Discussion – which makes it difficult to follow the argumentation. Improving the structure and avoiding repetition of information already presented in the Abstract and Results is recommended.
Reply 1: We appreciate the reviewer’s insightful comment. In response, we have thoroughly revised the Introduction, results and discussion to improve its clarity, coherence, and logical structure. We reorganized the content to present the background, problem statement, and objectives in a more structured and readable manner.
Comment 2: There are also some errors in the taxonomy section that should be corrected. For example, the Botryosphaeria complex was incorrectly assigned to the order Hypocreales, whereas it belongs to Botryosphaeriales (class Dothideomycetes). The term "Dematiaceous Coelomycetes" is outdated; it is recommended to use more current terminology, such as "pycnidial dark-spored fungi", or to refer to modern taxonomic groups (e.g., Phoma spp., Diaporthe spp.). Similarly, the use of "Deuteromycetes" in the Results section should be clarified, as it is no longer recognized as a valid taxonomic category.
Reply 2: update in the taxonomy was done in the results section.
Comment 3: In the section on physiological results, it would be helpful to indicate whether the differences in stomatal conductance were statistically significant, and to specify the number of plants per treatment group (n).
Replay 3: This was added in the legend of figure 8.
“Figure 8. Stomatal conductance and chlorophyll content in grapevine leaves across different treatments.
Physiological parameters were measured at 10 days post inoculation in grapevine plants subjected to four treatments: Control (Mock), Xylella fastidiosa subsp. fastidiosa (Xff), Sclerotinia sclerotiorum (Ss), and co-inoculation with X. fastidiosa + S. sclerotiorum (Xff/Ss).
Chlorophyll content (SPAD units) differed significantly among treatments (F = 2.01, p = 0.1664, df= 15). The highest mean value was observed in Xff/Ss (12.82 ± 4.69), followed by Xff (10.19 ± 3.01), Mock (9.26 ± 1.63), and Ss (7.73 ± 1.30). Tukey’s HSD test indicated that Xff/Ss was significantly different from Ss (p < 0.05). Stomatal conductance (mmol m⁻² s⁻¹) also varied significantly among treatments (F = 90.69, p < 0.0001, df= 15). The Ss treatment exhibited the highest conductance (451.1 ± 20.1), significantly higher than Xff/Ss (362.3 ± 8.1), Mock (199.4 ± 36.3), and Xff (139.0 ± 42.7), with all pairwise differences significant (p < 0.05).
Bars represent the mean ± standard deviation of four biological replicates per treatment. Different letters indicate statistically significant differences according to Tukey’s HSD test (p < 0.05).”
Comment 4: The manuscript also requires thorough language editing, both grammatically and stylistically. Many sentences are overly long or unnecessarily complex. There are also redundant statements – for instance, the role of Aureobasidium and Rhodotorula is described in similar terms several times. Care should be taken to ensure clarity in data presentation – for example, the notation "5:8 vs 8:8" needs to be clearly explained.
Reply 4: We thank the reviewer for this observation. We have thoroughly revised the manuscript for grammar, style, and sentence structure, with particular attention to simplifying overly long or complex sentences.
While we acknowledge the importance of avoiding redundancy, we respectfully maintain the mentions of Aureobasidium and Rhodotorula, as they appear in distinct sections in the manuscript (results and discussion). Moreover, we consider each little mention contributes to highlighting their ecological roles, or relevance to specific results (e.g., in biological control). Nevertheless, we have carefully reviewed the text to ensure that the information is not repetitive and that each reference provides added value to the narrative.
The data presentation has been clarified, and the notation "5:8 vs 8:8" has been removed and replace by: “Significant differences were found between Ss and Ss+Xff treatments for the average number of infective lesions on stems from the initial point of inoculation. The co-inoculation Ss+Xff shown a higher number of lesions than the others treatments”.
We trust that these changes enhance the clarity and quality of the manuscript.
Comment 7: Finally, the methodological and statistical sections should be improved. The manuscript lacks details on the number of replicates used in the greenhouse experiment, the p-values from statistical tests, and whether the assumptions of ANOVA were met. While the PCA was appropriately applied, it would benefit from a more detailed description of its parameters and interpretation.
Replay 7: These statistical details were clarified in the legend of each figure. We have clarified the number of replicates, included relevant p-values, F-values and degree of freedom, and provided details on the assumptions underlying the ANOVA in the revised figure legends and corresponding methods section. Additionally, we have expanded the description and interpretation of the PCA analysis, including parameter settings and the percentage of variance explained by each principal component, to enhance clarity and transparency.
Figure 5. Principal Component Analysis (PCA) biplot showing the distribution of treatments and disease severity variables in infected and mock-inoculated plants.
The biplot displays the first two principal components, F1 (87.52%) and F2 (10.62%), which together explain 98.14% of the total variance in the dataset. Treatments are represented as blue dots and include Xff, Xff-Ss, Ss, and Mock groups. Red vectors indicate the disease severity variables: number of lesions on stems, necrotic stem length (cm), and number of necrotic petioles per plant. Treatments inoculated with Xff-Ss are positively associated with higher values of all three disease parameters, as indicated by their position along the direction of the vectors. In contrast, Mock and Ss treatments cluster near the origin, reflecting minimal or no disease symptoms. The PCA suggests that the Xff-SS treatment caused the most severe symptoms, distinguishing it clearly from other treatments along PC1.
Comments 8: on the Quality of English Language
The manuscript is understandable, but would benefit from language editing to improve clarity and flow. Minor grammatical and stylistic revisions are recommended to enhance overall readability.
Replay 8: We appreciate the comment regarding the quality of the English language. The manuscript has been carefully revised to improve clarity, grammar, and overall flow. Minor stylistic and grammatical issues have been addressed throughout to enhance readability.

Round 2
Reviewer 2 Report
Comments and Suggestions for Authors
Thank you for your reply to my comments and revised manuscript
Sincerely,
Author Response
Reviewer Comment 1: Please provide a better description of the following results in the material and methods section, including the primer sequences and PCR conditions, not just citing the reference (Francis et al. 2006).
Response 1: We appreciate the reviewer’s observation. As requested, we have revised Section 4.2 of the Materials and Methods to include a detailed description of the primers and probes used for the detection of Xylella fastidiosa, as well as the complete qPCR reaction conditions and thermal cycling parameters. Specifically, we now provide the sequences of the primers HL5/HL6 and the TaqMan probe targeting the 16S rRNA gene, as described in Francis et al. (2008), along with the reaction volumes, reagent concentrations, and cycling conditions. These additions clarify the methodology and improve the reproducibility of our protocol.
4.2. qPCR assays to test the presence of Xylella fastidiosa (Xf)
DNA extraction from plant extracts was performed from leaf veins and petioles, and xylem sap using an EZNA HP Plant Mini kit (Omega-Biotek) following the manufacturer’s instructions, as described in the EPPO protocol (EPPO, 2016) [47]. The presence of Xff was assessed by real time PCR using two specific protocols with primers XF-F/XF-R and the TaqMan probe XF-P (Harper et al. 2010) and primers HL5/HL6 and the TaqMan probe HL-P(Francis et al. 2006) using an Applied Biosystems QuantStudio 3 Real-Time PCR System (Thermo Fisher Scientific, Waltham, Massachusetts, USA) [48,49]. Primers and qPCR conditions were as follows. For Harper qPCR, the target sequence is located in the gene coding for the 16S rRNA processing RimR protein. Sequences were as follows: forward primer, XF-F, 5’-CACGGCTGGTAACGGAAGA-3’; reverse primer, XF-R, 5’-GGGTTGCGTGGTGAAATCAAG-3’; and sequence of probe, XF-P, 5’-6-FAM-TCG CATCCCGTGGCTCAGTCC-BHQ-1-3’. PCR conditions: pre-incubation of 50 °C for 2 min, an initial denaturation of 95°C for 10 min, followed by 40 cycles of 95°C for 10 s and 62°C for 40 s. Heating ramp speed: 1.6 °C/s. For Francis qPCR, the target sequence is a conserved hypothetical protein HL gene. The sequence primers are as follows: forward primer HL5: 5’-AAGGCAATAAACGCGCACTA-3’; reverse primer HL6: 5’-GGTTTTGCTGACTGGCAACA-3’; the probe sequence is 5’-6-FAM-TGGCAGGCAGCAACGATACGGCT-BHQ-1-3’. Pre-incubation (UNG step) at 50°C for 2 min, initial denaturation at 95°C for 10 min, followed by 45 cycles of 95°C for 15 s and 60°C for 60 s. Heating ramp speed: 1.6 °C/s. All samples were analyzed in triplicate including the positive and negatives controls.
Reviewer Comment 2: In addition, please include a table describing the morphological traits used for the fungal characterization of the strains, at least for those mentioned in Figure 1.
Response 2: We thank the reviewer for this valuable suggestion. In response, we have added a new table (now Supplementary Table 1) summarizing the morphological traits used to characterize the fungal strains mentioned in Figure 1. This table includes macroscopic features (e.g., colony color, texture) and microscopic traits (e.g., hyphal structures, conidia/spore morphology, presence of sexual/asexual fruiting bodies), following standard mycological guidelines. We believe this addition provides greater clarity and completeness to the characterization of the fungal isolates.
Supplementary Table 1. Morphological traits used for the fungal characterization of the studied genera and fungal complexes shown in Fig 1.
|
|
Key Diagnostic Features |
|
|
Fungal Group |
Macroscopic Traits (Culture) |
Microscopic Traits |
|
Phoma complex |
Slow-growing colonies, grayish to olivaceous, sometimes producing pinkish pigment. |
Pycnidia with ostiole, globose to flask-shaped, producing slim conidial masses; hyaline, aseptate conidia. |
|
Phomopsis / Diaporthe complex |
Creamy to gray colonies, often with concentric zonation or black pycnidia. |
Two conidial types; Alpha conidia: ellipsoid, aseptate; Beta conidia: filiform, curved or straight. |
|
Alternaria alternata |
Fast-growing, dark olive to blackish colonies, often with zonation. |
Conidia large (20–63 µm long × 7–18 µm wide) with apical beak, muriform (both transverse and longitudinal septa), often in chains.; typical dark pigmentation. |
|
Cladosporium spp. |
Velvety to powdery colonies, olive-gray to greenish-brown in color. |
Conidia in branching chains, with scars (coronate hilum); conidiophores branched |
|
Aureobasidium pullulans |
Colonies initially yeast-like cells, turning dark and leathery with age. |
Dimorphic nature; production of melanin; presence of both yeast and filamentous forms and dark-walled chlamydospores may be present. Budding blastoconidia produced in large numbers; hyaline, oval to ellipsoidal, unicellular; size: 4–10 × 2–4 µm. |
|
Rhodotorula mucilaginosa |
Yeast-like growth with bright pink to coral colonies; smooth, mucoid, shiny |
Oval to round yeast cells, reproducing by budding; absence of pseudohyphae or mycelium. |
|
Penicillium spp. |
Rapidly growing, velvety to powdery colonies, typically blue-green with a white margin. |
Brush-like conidiophores with branched metulae and phialides; conidia produced in chains. |
|
Botryosphaeria complex |
Dark pigmented, slow- to moderate-growing colonies. |
Pycnidia or pseudothecia; fusiform, aseptate or septate conidia, occasionally with appendages. Asexual and sexual structures; dark, thick-walled pycnidia characteristic of the complex. |
|
Phaeoacremonium / Phaeomoniella spp. |
Pale brown to dark pigmented colonies; often slow-growing. |
Pigmented hyphae; slimy phialidic conidia; presence of type-specific phialides. phialidic conidiogenesis |
|
Aspergillus spp. |
Powdery colonies, green or black depending on species; rapid growth. |
Conidiophores terminate in vesicle; phialides in uniseriate or biseriate arrangements; conidia in chains. |
|
Yeast-like fungi (general) |
Creamy, smooth, shiny colonies; white to pink or tan in color colony pigmentation |
Oval or ellipsoidal yeast cells; reproduce by budding; pseudohyphae presence/absence |
|
Sclerotinia sclerotiorum |
Rapid growth; white surface, cottony to fluffy aerial mycelium that covers the entire plate surface and black spherical to elongated sclerotia of 6.4 (2 to 10) mm. |
Numerous black, irregularly shaped sclerotia form on colony surface and within agar, variable in size (2–10 mm). Applanate or embedded in agar, often peripheral or scattered across the colony. Lack of conidia, typically non-conidial; identification relies on mycelial and sclerotial traits. |
Culture characteristics were assessed on PDA 2% medium or Sabouraud medium for 3-7 days in a growth chamber at 25°C, 12 h of light and 12 h of darkness. Microscopic structures were observed in slide cultures stained with lactophenol cotton blue.
